# Potential Roles and Future Perspectives of Chitinase 3-like 1 in Macrophage Polarization and the Development of Diseases

**DOI:** 10.3390/ijms242216149

**Published:** 2023-11-09

**Authors:** Hailong Zhao, Mingdong Huang, Longguang Jiang

**Affiliations:** College of Chemistry, Fuzhou University, Fuzhou 350116, China; 211320349@fzu.edu.cn

**Keywords:** CHI3L1, macrophage polarization, inhibitors, diseases, cancers

## Abstract

Chitinase-3-like protein 1 (CHI3L1), a chitinase-like protein family member, is a secreted glycoprotein that mediates macrophage polarization, inflammation, apoptosis, angiogenesis, and carcinogenesis. Abnormal CHI3L1 expression has been associated with multiple metabolic and neurological disorders, including diabetes, atherosclerosis, and Alzheimer’s disease. Aberrant CHI3L1 expression is also reportedly associated with tumor migration and metastasis, as well as contributions to immune escape, playing important roles in tumor progression. However, the physiological and pathophysiological roles of CHI3L1 in the development of metabolic and neurodegenerative diseases and cancer remain unclear. Understanding the polarization relationship between CHI3L1 and macrophages is crucial for disease progression. Recent research has uncovered the complex mechanisms of CHI3L1 in different diseases, highlighting its close association with macrophage functional polarization. In this article, we review recent findings regarding the various disease types and summarize the relationship between macrophages and CHI3L1. Furthermore, this article also provides a brief overview of the various mechanisms and inhibitors employed to inhibit CHI3L1 and disrupt its interaction with receptors. These endeavors highlight the pivotal roles of CHI3L1 and suggest therapeutic approaches targeting CHI3L1 in the development of metabolic diseases, neurodegenerative diseases, and cancers.

## 1. Introduction

Chitinases and chitinase-like proteins (CLPs) are proteins that are widely present in mammals and humans. The roles of chitinases and CLPs have been demonstrated in the fields of plant and microbial immunity [1]. Host-derived chitinases cleave chitin to prevent invasion by chitin-carrying pathogens. Compared to plants, mammals lack endogenous chitin or chitin synthase. However, chitinase and chitinase-like proteins are also endogenously expressed in the lungs and other organs of animals [2,3,4,5]. CLPs are involved in the mediation of many diseases characterized by chronic inflammation and tissue remodeling. Chitinase and CLPs are mainly expressed and secreted by macrophages and are closely related to M2 macrophage activation [6]. Therefore, understanding the physiological functions of chitin or chitinase-like proteins is crucial for disease prevention and treatment.

CHI3L1 (chitin-3-like 1), also known as YKL-40 in humans, is a member of the 18-glycoside hydrolase family composed of 383 amino acids. CHI3L1 has a molecular weight of 40 KDa and no hydrolase activity due to mutations in two critical active residues, but it retains a high affinity for chitin. A comparison of CHI3L1 sequences from different species shows a high degree of conservation (shown in Figure 1a). This indicates that studying the physiological functions and activities of CHI3L1 in animals can be strongly correlated with relevant physiological activities in humans, which has significant clinical research significance.

CHI3L1 is secreted and expressed by various cells, including macrophages [7], neutrophils [5], tumor cells [8], and vascular smooth muscle cells [9]. It plays a significant role in cell regeneration, proliferation [10], migration, tissue remodeling [11], and angiogenesis [12]. Levels of CHI3L1 mRNA and protein are significantly elevated in inflammatory diseases [8,13], cancer [14], and degenerative diseases [15] and are closely associated with patient survival and poor prognosis. In recent years, extensive research has led to the recognition of CHI3L1 as a widely used biomarker and drug target.

The crystal structure of CHI3L1 reveals two domains: an eight-stranded β/α-barrel domain and a second domain consisting of six antiparallel β-strands and an α-helix (shown in Figure 1b). Structural analysis identifies two distinct heparin-binding domains in CHI3L1, each containing positively charged arginine and lysine residues that interact with heparin through electrostatic interactions [16]. However, a study investigating the binding of heparin and CHI3L1 presents contrasting findings. By synthesizing short peptides with mutated KR-rich domains and measuring their binding strength to heparin using ELISA [17], it is demonstrated that the actual binding site between CHI3L1 and heparin is located in the C-terminal KR-rich domain rather than the theoretically proposed binding motif (RRDK) (shown in Figure 1c). Besides heparin, CHI3L1 also interacts with various proteins and small molecules, such as IL-13Rα2 [18], CD44 [19], TMEM219 [20], Galectin-3 [21], and chitin [16]. Due to the existence of different receptors for CHI3L1, it exerts its physiological functions through diverse signaling pathways. Understanding these mechanisms is crucial in comprehending the role of CHI3L1 in different diseases.

In recent studies on COVID-19, significant discoveries have emerged. This disease, caused by SARS-CoV-2 (SC2), has been observed to be more severe and widespread in elderly individuals and those with comorbidities. Interestingly, research conducted by Kamle’s team indicates that CHI3L1, which is induced during aging and comorbidity, acts as a potent stimulator for the SARS-CoV-2 receptor angiotensin-converting enzyme 2 (ACE2) and spike protein-priming proteases (SPP) [22]. These findings shed light on the role of CHI3L1 in COVID-19 pathogenesis. Moreover, further research has found that CHI3L1 enhances SC2 infection by using the CHI3L1 axis, playing a critical role in the pathogenesis of COVID-19, and is an attractive therapeutic target. Kamle et al. [23] demonstrated that CHI3L1 increased the expression of ACE2 and SPP in epithelial cells by studying typical delta (δ)- and omicron (ο)-variant mutations of the virus, leading to the uptake of pseudo-viruses expressing alpha, beta, gamma, delta, or omicron S proteins by epithelial cells, thereby enhancing their infection. And anti-CHI3L1 inhibitors and Kanamycin partially inhibit the infection of epithelial cells by these variant-of-concern (VOC) pseudo-viruses, once again proving that CHI3L1 is a common and VOC-independent therapeutic target in COVID-19.

A large amount of data shows that research on CHI3L1 has become increasingly important, as it is involved in disease progression, inflammatory responses, fungal infections, cell activation, and other physiological activities. Whether as a biomarker of disease or a therapeutic target for related diseases, CHI3L1 is an extremely important protein. Based on this, this review summarizes recent studies on CHI3L1 to illustrate how CHI3L1 works in different diseases. The aim is to find general rules and provide new ideas for future CHI3L1 research.

## 2. Correlation between CHI3L1 and Macrophages

Originally, macrophages were defined as mobile cells with phagocytic activities. Most macrophages originate from monocytes, migrate to the circulatory system, and then move to peripheral tissues as needed. However, the exact function of macrophages has not been fully studied [24]. In some tissues, macrophages have specialized phenotypes, such as microglia in the central nervous system, alveolar macrophages in the lungs, and Kupffer cells in the liver. These different phenotypes of macrophages play important roles in maintaining tissue homeostasis [25]. At the same time, as important immune cells involved in innate immunity, macrophages have significant heterogeneity and polarization [26].

Macrophages have two common polarization phenotypes: the classically activated M1 phenotype and the alternatively activated M2 phenotype. M1 macrophages can be differentiated by LPS, IFN-γ, and CSF2 stimulation [27], while M2 macrophages can be differentiated by IL-4, IL-13, TGF-β, TNF-α, and glucocorticoids [26,28,29,30]. Macrophage polarization is temporary and malleable, and macrophages can quickly change their phenotype to adapt to changes in the surrounding tissue microenvironment. In other words, macrophages can be transformed from the M1 phenotype to the M2 phenotype under certain conditions, while M2 macrophages can reprogram various genes expressed by M1 macrophages by using reagents that increase IL-10 levels [31,32].

The close relationship between tumor-associated macrophages (TAMs) and CHI3L1 has been demonstrated in many diseases, especially during inflammation. Macrophages are the precursor of TAMs, and research on their immune regulation function helps in further understanding the role played by TAMs in diseases. Numerous research findings have consistently demonstrated significant physiological associations between macrophages and CHI3L1 across various diseases, establishing their correlation with the two classical phenotypes of macrophage differentiation. In the following discussion, we delve into the interplay between the M1 and M2 phenotypes alongside CHI3L1 while also providing an overview of the connection between CHI3L1 and macrophages in select diseases.

In a mouse model of breast cancer, co-culturing 4T1 cells with RAW264.7 cells in vitro resulted in a significant increase in CHI3L1, LCN2, and MMP-9 levels in serum, which promoted tumor metastasis [33]. The same results were observed in a mouse mammary ductal carcinoma model. In addition, silencing the CHI3L1 gene or treating it with recombinant CHI3L1 (rCHI3L1) can reduce the level of IFN-γ cytokines produced by M2 macrophages [34]. Spleen macrophages carrying breast cancer tumors secrete higher levels of proinflammatory mediators CCL2, CXCL2, MMP-9, and CHI3L1 stimulating this increased secretion [35]. Treatment with rCHI3L1 also enhances the expression of CCL2, CXCL2, and MMP-9 in mouse stromal and alveolar macrophages after LPS treatment, consistent with the aforementioned results that CHI3L1 stimulates macrophages to produce IL-8 (mouse CXCL2 homolog), MCP-1 (CCL2), and MMP-9 [34,36].

CHI3L1 is also specifically upregulated in cancer-associated fibroblasts (CAFs), and CAF-derived CHI3L1 appears to be unique in that it can reprogram TAMs into an M2-like phenotype, which is associated with tumor progression. Knockdown of CHI3L1 in CAFs results in a decrease in M2-like macrophages, an increase in CD8+ T cells, and a shift in CD4+ T cells towards the TH1 phenotype. This indicates that CHI3L1 from CAFs affects macrophage pathological phenotypes by affecting TAMs and T cells [33]. The promotion of CCL2, CXCL2, and MMP-9 by CHI3L1 shows the opposite effect to its downregulation of IFN-γ produced by M1 macrophages but ultimately leads to the same result: promoting macrophage differentiation into the M2 phenotype.

The promotion of M2 macrophage differentiation has also been found in other diseases. Lung metastasis studies have found that CHI3L1-KO mice have reduced Th2 inflammation, while overexpression of CHI3L1 reverses this phenotype and is expressed in activated T cells and Th2 cells, regulating Th1 and Th2 differentiation by increasing IFN-γ signaling pathways. CHI3L1 enhances type 2 immune responses, stimulates M2 macrophage differentiation and the formation of TGF-β1, and regulates melanoma and breast cancer metastasis through the Sema7a/CHI3L1/IL-13Rα2 axis [37]. In mouse models of A549 lung cancer, the administration of anti-CHI3L1 antibodies has been observed to modulate the tumor microenvironment. These antibodies exert their effects by suppressing STAT6-dependent PLG signaling, consequently leading to a reduction in M2 polarization [38]. M2 macrophage differentiation, to some extent, is advantageous, and the relationship between CHI3L1 and macrophage polarization may bring a new therapeutic approach.

Understanding the differentiation mechanism of macrophages in atherosclerosis is crucial for comprehending the associated regulatory mechanisms. Jung et al. found that CHI3L1 promotes the expression of PPARδ in THP-1 and HUVECs [39]. Furthermore, it inhibits NF-κB phosphorylation and the secretion of proinflammatory factors such as TNF-α and MCP-1 induced by LPS during atherosclerosis. Acting as a stimulus for M1 macrophages, CHI3L1′s ability to suppress TNF-α secretion is thought to mediate macrophage differentiation and contribute to atherosclerosis. However, further investigation is required to determine whether the attenuation of Th1 immune responses results in the dominance of Th2 responses in CHI3L1-mediated macrophage differentiation mechanisms in atherosclerosis. This potential explanation may shed light on why CHI3L1 levels increase with disease progression, but additional research is needed to confirm this theory [40].

The relationship between CHI3L1 and macrophage differentiation in different diseases is summarized in Table 1. However, relevant reports are not limited to the diseases mentioned above. For example, in CHI3L1-KO mice with occlusion of the middle cerebral artery injury, the absence of CHI3L1 accelerates stroke development through the activation of STAT6-dependent M2 microglia [41]. In choroidal neovascularization (CNV) development, CHI3L1 can regulate M2 differentiation-mediated angiogenesis via VEGFA-dependent or independent pathways [42]. CHI3L1 promotes food allergies through Th2 immune responses and the combination of MAPK/ERK and M2 macrophage polarization-mediated Th2 immune responses and PI3K/AKT signaling pathways [43]. Hyaluronic acid (HA) may promote macrophage recruitment and M2 polarization through IL-1/CHI3L1 and TGF-b/CHI3L1 axes [44], in which CHI3L1 plays a key role in Th2 inflammatory response, M2 macrophage activation, and skin barrier function in the development of specific dermatitis [45]. Monocyte-derived cells strongly express CHI3L1 under the stimulation of Th2 cytokines IL-4 and TGF-β [46]. miR-24 targets the CHI3L1 gene, promotes M1 macrophage polarization, reduces M2 macrophage polarization, etc. [47]. Although the mechanism of CHI3L1′s impact on these diseases may not have been fully studied, research data certainly indicate a close correlation between CHI3L1 and macrophage differentiation. The infiltration, activation, and polarization of macrophages play different roles in different diseases. By summarizing the relationship between CHI3L1 and macrophage polarization or how it affects polarization in different diseases, the results show that most CHI3L1 is secreted by M2-like macrophages and promotes M2-like polarization of macrophages in most diseases.

## 3. Therapeutic Approaches of CHI3L1 for Treatment of Diseases

As further research is conducted on the function, structure, and physiological effects of CHI3L1 protein, scientists have discovered and identified many inhibitors, including antibodies, small molecules, miRNA, and others, as shown in Table 2. Fascinatingly, research findings about various CHI3L1 inhibitors have once again underscored the significant association between CHI3L1 and lung cancer, breast cancer, as well as neurological diseases. Notably, the inhibitory effects of small molecules on CHI3L1 predominantly involve regulating the functional polarization of macrophages or impeding the binding of CHI3L1 to receptors, thereby contributing to the amelioration of cancer progression. The substantial advancements achieved through CHI3L1 inhibition have significantly improved disease progression while demonstrating favorable biological safety and holding tremendous promise for future application.

## 4. CHI3L1 and Different Diseases

### 4.1. Correlation between CHI3L1 and Diabetes and Atherosclerosis

Diabetes is a metabolic disease characterized by high blood glucose levels due to dysfunction in the body’s glucose-handling mechanism, often caused by insulin secretion defects or impaired biological function, or both, and is generally classified into type 1 and type 2 diabetes (T1/2D) [61]. Elevated levels of CHI3L1 have been found in the plasma of both types of diabetes, and CHI3L1 levels are closely related to insulin resistance in T2D [62,63,64]. The increase in CHI3L1 serum levels in diabetic patients suggests its potential as a relevant marker, and further studies have found that CHI3L1 not only stimulates the growth and proliferation of fibroblasts but also participates in the degradation or breakdown process of connective tissue inflammation. In T1D patients, elevated plasma CHI3L1 levels are positively correlated with proteinuria, indicating that CHI3L1 plays a role in microvascular disease caused by renal vascular damage.

Inflammation is involved in all aspects of human immunity, metabolism, and disease progression. Subclinical systemic inflammation exists in T2D and participates in the pathogenesis of all stages of atherosclerosis [65]. Plasma CHI3L1 levels in T2D patients are positively correlated with insulin resistance and blood lipid abnormalities, and CHI3L1 levels were 50% higher than the median in a control group. This indicates that using CHI3L1 as a marker for T2D patients is very practical [62]. Especially with an increased level of CHI3L1 in the plasma of T2D patients, the secretion of IL-6 and tumor necrosis factor-alpha (TNF-α) is enhanced by cytokine IL-18, which is closely related to insulin [66]. This suggests a possible functional relationship between IL-18, IL-6, TNF, and CHI3L1, although there is no specific evidence to prove it. The correlation between CHI3L1 content in plasma and IL-6 has been partially validated [67].

Atherosclerosis is a common complication of diabetes and an important cause of coronary artery disease and stroke in clinical practice. In the early stage of atherosclerotic disease, chronic inflammation usually occurs, and as the disease progresses, plaques composed of lipids, necrotic cores, calcification areas, inflammatory smooth muscle cells, endothelial cells, immune cells, and foam cells form and grow on the vascular wall, leading to narrowing or rupture of the vascular channel. Severe cases can result in bleeding due to vascular wall rupture. Plaque erosion and rupture are also considered the main causes of acute vascular events, such as acute myocardial infarction, stroke, and sudden death [68,69,70,71]. In the inflammatory process, blood monocytes migrate from the blood to lymphoid and non-lymphoid tissues in response to signals from tissues such as infection or tissue damage [72]. They engulf other cells and toxic molecules such as oxidized LDL (oxLDL), produce inflammatory cytokines, and can differentiate into inflammatory dendritic cells (DCs), macrophages, or foam cells [73,74]. Previous studies have shown that CHI3L1 expression in different subpopulations of macrophages in atherosclerotic plaques is highly upregulated, and the expression levels of CD36, CD14, and CD18 are significantly increased in monocytes of diabetic patients. Differentiation and maturation of CD14+ monocytes into CD16+ macrophages are also accompanied by CHI3L1 expression [75].

The relationship between CHI3L1 and atherosclerosis is closely related to macrophage activation, and the enhancement of TNF-α also promotes the expression of CHI3L1 in macrophages. Another study found that the activation of TLR2 and TLR4 further increased the release of CHI3L1 in THP-1 monocytes activated by TNF-α [76]. CHI3L1 mRNA expression is highly upregulated in different subpopulations of macrophages in atherosclerotic plaques. The formation of plaques involves the infiltration of monocytes into the subendothelial space of the vascular wall, followed by lipid accumulation in activated macrophages. Macrophages infiltrating deep into lesions have higher mRNA expression of CHI3L1 than other lesion sites, and this is expressed highly in early-stage macrophages of atherosclerotic lesions. CHI3L1 may inhibit macrophage apoptosis by upregulating the expression of the apoptotic inhibitor Aven and inhibiting the activation of the apoptotic initiator caspase-9. This inhibition of macrophage apoptosis leads to impaired programmed cell removal (PrCR) in plaques, and macrophages that should undergo apoptosis and be removed by phagocytosis gradually accumulate, eventually enlarging plaques and exacerbating the formation of early-stage atherosclerosis [77]. Yue et al. used xCell analysis to obtain immune cell enrichment from the GSE41571 database and found that, compared to stable plaques, ruptured plaques had higher infiltration of plasma B cells, bone marrow-activated dendritic cells, and macrophages as well as higher infiltration of M1 and M2 macrophages [78].

In addition, studies have found that CHI3L1 is not only upregulated in more vulnerable plaques but also significantly correlated with inflammatory markers and the loss of stable differentiated SMC content [79]. CHI3L1 enhances endothelial cell migration and angiogenesis through its receptor IL-13Rα2-dependent pathway and activates downstream signaling molecules AKT and ERK [80]. IL-13Rα2, one of CHI3L1′s receptor proteins, participates in and enhances the migratory effect of CHI3L1 on EC cells, and two other studies have obtained similar results [81,82]. The upregulation of CHI3L1 may represent a compensatory response aimed at preventing SMC ‘dedifferentiation’ and inflammation [79], which often occurs in conjunction with macrophage differentiation, consistent with the relationship between CHI3L1 and macrophages mentioned above.

### 4.2. Correlation between CHI3L1 and Liver Diseases

The liver is an important organ in the human body, responsible for detoxification, metabolism, promoting blood circulation, immune defense, and other important physiological functions. Viral, bacterial, and parasitic infections, improper diet, alcohol consumption, stones, and other diseases can cause lesions in the liver, leading to different types of liver diseases. Many scholars have already discovered changes in the content of CHI3L1 in liver injury, liver cancer, liver fibrosis, cirrhosis, and other diseases [83,84,85,86,87,88,89].

Recently, a study on obstructive sleep apnea–hypopnea syndrome (OSA) pointed out that, as a possible precursor to liver fibrosis, if OSA is detected and diagnosed, it can prevent liver fibrosis to some extent [90]. By studying the levels of five serum liver fibrosis markers in OSA patients, a novel marker—CHI3L1—was found, and detecting CHI3L1 content in serum is safer and faster than detecting other markers. Moreover, compared to those in the normal population, levels of hyaluronic acid, collagen type IV, and CHI3L1 in the serum of OSA patients were significantly increased, indicating that detecting the serum level of CHI3L1 to determine the progression of OSA and prevent fibrosis is possible [91]. Early diagnosis and continuous monitoring of liver fibrosis are crucial for the clinical treatment of chronic diseases and prognosis evaluation. Using serum CHI3L1 content alone as an indicator to predict the degree of fibrosis is feasible, but the level of CHI3L1 in the liver gradually increases with normal aging, which is unrelated to the existence of liver fibrosis disease (shown in Figure 2).

A study indicated that the accuracy of joint diagnosis of fibrosis by using serum CHI3L1 levels and HA levels was higher than that of using serum CHI3L1 levels alone [84,91,92]. In the process of liver fibrosis, CHI3L1 promotes the proliferation and activation of hepatic stellate cells by stimulating the production of COL1A1 and ACTA2. This process is independent of TGF-β1 regulation. Additionally, CHI3L1 plays a crucial role in liver fibrosis by directly affecting hematopoietic stem cells [93,94], which is corroborated by the results of Masaaki et al. [95], who found that CHI3L1 deficiency significantly reduces the level of liver TGF-α but does not affect the level of TGF-β and that CHI3L1 is involved in the apoptosis of liver macrophages by mediating Fas expression and the Akt signaling pathway. The above results suggest that CHI3L1 mainly inhibits the apoptosis of M1-like liver macrophages rather than their M2 counterparts in liver fibrosis, which requires further research to explain.

In patients with acute liver injury (ALI), the level of CHI3L1 is negatively correlated with the level of ALI [89]. Diseases caused by factors such as oxidative stress, excessive inflammatory response, and mitochondrial damage usually involve inflammation, resulting in liver cell damage and apoptosis, thereby affecting normal liver function [96,97]. Acetaminophen (APAP) is a common analgesic used to relieve pain and reduce fever. Liver damage caused by APAP is called APAP-induced liver injury (AILI). Overuse of APAP is the main cause of acute liver damage in the United States and can lead to a large accumulation of platelets in the liver. Shan et al. found that CHI3L1 recruits platelets through its receptor CD44 on macrophages, thus promoting AILI [98]. In the TAA-induced ALI mouse model, rCHI3L1 inhibits the differentiation of IFN-γ+ Th1 cells through the STAT3 signaling pathway, thereby suppressing IFN-γ secretion and improving Th1 cell-mediated inflammatory response in the liver, providing a possible target for ALI treatment [98,99]. Recent research has obtained highly specific humanized CHI3L1 monoclonal antibodies through single-memory B-cell culture and achieved significant results in treating AILI, indicating the feasibility of CHI3L1 as a target for ALI treatment [100].

Further research on the liver and CHI3L1 shows that the level of CHI3L1 in the liver is positively correlated with the level of CHI3L1 in circulation, and liver cells are the main source of CHI3L1 in the liver. The proinflammatory cytokine IL-6 significantly increases CHI3L1 expression in primary liver cells independently of TGF-β1 [84,101,102,103]. In vivo studies have shown that CHI3L1 enhances caspase-3 and cleaved-caspase-3 expression levels in liver cells through the PAR2/JNK signaling pathway, thus exhibiting a significant pro-apoptotic effect on liver cells. The JNK pathway is involved in cell apoptosis in the liver and, together with CHI3L1, it forms the CHI3L1/PAR2/MAPK/JNK/caspase-3 liver cell apoptosis pathway [104]. The high expression of CHI3L1 in liver cells, combined with its pro-apoptotic effect on liver cells, form a negative feedback loop.

In addition, a review of macrophages and their role in liver disease pointed out that M1/M2 macrophage polarization plays an important role in the progression of liver diseases. Maintaining macrophage polarization of a certain phenotype also depends on different types of liver diseases. For example, in ALI and alcoholic liver disease (ALD), maintaining M2 macrophage polarization is more conducive to inflammation and disease relief. In HBV, HCV infection, and hepatocellular carcinoma (HCC), the M1 macrophage phenotype is more conducive to disease relief and tumor development. Different phenotypes of macrophages have different effects on different types of liver fibrosis during progression [26]. CHI3L1 is mainly derived from M1 macrophages and has higher expression levels in M1 macrophages than in other subgroups [95]. The specific physiological effects of CHI3L1 in various liver diseases have not been thoroughly studied. However, due to the complexity and variability of liver diseases and the different effects of M1/2 polarization on disease progression, even in the same disease, the influence of macrophage polarization may vary depending on different inducers. CHI3L1 and macrophage polarization certainly play an important role in the progression of liver diseases, making it particularly important to study the physiological function of CHI3L1 in liver diseases.

### 4.3. Correlation between CHI3L1 and Neurological Diseases

Central nervous system (CNS) diseases, also known as neurodegenerative diseases, are characterized by cognitive decline, motor disorders, behavioral changes, and other clinical features [105]. Alzheimer’s disease (AD) is a CNS disease, and the progression of AD generally includes three stages—preclinical (asymptomatic), mild cognitive impairment (MCI), and resulting dementia. AD is caused by various factors, such as aging, smoking, obesity, diabetes, etc. [106,107]. The pathological features of AD are mainly located in neurofibrillary tangles (NFTs) inside neuronal cells and extracellular amyloid plaques (Aβ), the former composed of different forms of phosphorylated Tau protein and the latter composed of Aβ deposits throughout the brain [108].

As a persistent and difficult-to-recover neurodegenerative disease, early prevention and treatment of AD, as well as late-stage treatment, are very important. The reliable diagnosis of AD generally involves detecting amyloid plaques and NFTs composed of pathological deposits of Tau protein in neuronal tissue, but this type of tissue detection is difficult to achieve.

Modern medical research has focused on measuring biomarkers in various body fluids to infer the progression of different diseases, such as cerebrospinal fluid (CSF) and blood [109,110]. Several studies have found that the level of CHI3L1 in AD patients significantly increases and further increases with disease progression, providing a new, minimally invasive, and easily obtained AD biomarker [111,112,113]. Rosen et al. found that the level of CHI3L1 in the CSF of AD patients was significantly higher (77%) than that of normal individuals [114,115]. At the same time, another related study indicated that CSF CHI3L1 is not a pre-biomarker for early symptoms of AD, which makes measuring AD progression by measuring CHI3L1 levels in CSF seem unreliable. However, most researchers still believe that using plasma CHI3L1 as a biomarker is very promising, while a better method is to combine other markers to determine AD progression to obtain more accurate results [15,114,116].

As mentioned before, the main cause of AD is the presence of chronic brain inflammation. A report about the brains of AD patients and normal individuals found that the significant difference in CHI3L1 can be attributed to an imbalance in the cell activation spectrum, with increased neuronal activity in the low-CHI3L1-expression group (LCEG) and enhanced inflammatory processes mediated by microglia activation in the high-CHI3L1-expression group (HCEG), suggesting that CHI3L1 may activate microglia to promote inflammation and AD progression and may be related to gender [117]. This is consistent with the conclusion of another study: Brian et al. used animal models to further reveal that CHI3L1 knockout changes neurogenic inflammation responses in animal models while promoting astrocytes and microglia to engulf Aβ, reducing the formation of amyloid plaques, indicating that CHI3L1 knockout has potential protective effects in AD but enhances LPS-induced inflammatory responses and does not affect amyloid plaque deposition, suggesting that CHI3L1 is a potential therapeutic target for limiting plaque accumulation, optimizing plaque engulfment response, and slowing down AD progression, but it is destructive in acute inflammatory environments [118].

Glial cells play an important role in the progression of AD, and glial activation in AD is believed to have both advantages and disadvantages. On the one hand, activated glia can engulf Aβ and Tau proteins to prevent protein damage, but on the other hand, excessive inflammatory activation can accelerate plaque accumulation and synaptic loss [119]. In addition to regulating glial cell activation, CHI3L1 also regulates the release of proinflammatory cytokines in neuroglia. Increased expression of CHI3L1 has been found in post-mortem tissue samples from sporadic AD patients aged 70–80 and is further enhanced by systemic infection. Strangely, no significant changes in CHI3L1 mRNA were observed in young AD patients with severe symptoms, which makes us wonder if the increase in CHI3L1 expression in AD requires age-related synergy or infection [13].

The high expression of CHI3L1 in nearly all brain disorders may be attributed to the activation of astrocytes during disease states, leading to gliosis and exacerbating disease progression [120]. It remains unclear whether CHI3L1 functions as a neuroinflammatory signaling molecule that triggers receptor systems in brain cells and regulates inflammatory responses in a cell-type-specific manner [121]. Moreover, given the multitude of receptors and signaling pathways associated with CHI3L1, it is equally imperative to investigate whether CHI3L1 acts upon established receptors and signaling pathways in the brain or potentially novel receptors to comprehend its involvement in Alzheimer’s disease or the advancement of brain disorders [122].

Fungal infections have been linked to cancer, including 35 types of cancer, mostly located intracellularly [123,124]. In 2014, researchers discovered fungal biomolecules in the brains of Alzheimer’s disease patients for the first time [125]. Treatment with antifungal medication reversed dementia in many patients. Fungal infections typically induce inflammatory reactions and vascular modifications, similar to the slow progression of AD. Fungal infections can induce the Th1 signaling pathway, producing TNF, IFN-β, IL-1, IL-6, and IL-12, and can also induce Th2 responses. CHI3L1, as a member of the chitinase family, plays an important role in fungal infection models. In a model of corneal candidiasis, CHI3L1 is primarily expressed in epithelial cells and exerts protective effects against fungal infection by influencing the expression of anti-inflammatory factors and chemokines through an IL-13Rα-2-dependent mechanism [126]. CHI3L1 mediates cell activation in inflammation progression, which may be closely related to its antimicrobial effects in the brains of AD patients. In the invasive pulmonary aspergillosis (IPA) model, pentoxifylline (PTX) significantly inhibits CHI3L1 expression in non-neutropenic IPA mice (HC-IPA) [127], suggesting that CHI3L1 may also be a potential target for fungal infection and inflammation in the brains of AD patients, which is worth further exploration by researchers.

Regarding CHI3L1-related treatment methods, some researchers have achieved certain results. A team in Korea screened for CHI3L1 inhibitors in a drug library and discovered K284-6111, which can inhibit CHI3L1 to suppress the ERK and NF-κB signaling pathways. Another inhibitor, G721-0282, can also reduce neuroinflammation induced by chronic unpredictable mild stress (CUMS) and alleviate anxiety behavior characteristics [57,128]. This is similar to previous research, where small molecules inhibited NF-κB pathway activation, reducing the expression of genes such as iNOS and decreasing the accumulation of amyloid plaques (shown in Figure 3). The success of early experiments on CHI3L1 inhibitors once again suggests that targeting CHI3L1 as a therapeutic target is feasible, but there is still a long way to go.

## 5. Correlation between CHI3L1 and Cancers

The close connection between CHI3L1 and cancer has been proposed by numerous researchers and is now well established. In many types of cancer, the level of CHI3L1 is related to disease progression and prognosis, with overexpression of CHI3L1 generally associated with poor outcomes. Studies have shown that the spread and metastasis of tumors often accompany the expression of CHI3L1 from macrophages [14,38,129,130,131,132]. CHI3L1 participates in cancer progression through different pathways, such as enhancing the production of proinflammatory/pro-tumor angiogenic factors to aid tumor spread or regulating signaling pathways in tumor progression [34,35,129]. In addition to directly affecting cancer progression, CHI3L1 also acts as an intermediate protein involved in the regulation of other proteins in cancer [59,133,134,135,136].

Most of the CHI3L1 involved in cancer progression and regulation comes from macrophages, a group of cells that play an important immune role in the body. Macrophages participate in the metastasis and proliferation of cancer cells by secreting cytokines. Tumor-associated macrophages play a crucial role in cancer progression. Different stimuli can cause macrophages to differentiate into various phenotypes, and TAMs infiltrating into cancer sites have different physiological functions than normal macrophages. Researchers have found that TAMs in breast tumors differ in phenotype and function from macrophages in normal breast tissue [137]. The functional polarization of macrophages plays a crucial role in resisting cancer cells and controlling diseases.

As mentioned earlier, the relationship between different subtypes of macrophages and CHI3L1 has been discussed. However, due to the variability of inducers, the complexity of the cancer environment, and the variability of the disease, it is difficult to summarize the role of the CHI3L1 protein in different cancers. Therefore, the following subsections elaborate and summarize several studies involving CHI3L1 and various cancers and macrophages.

### 5.1. CHI3L1 and Breast Cancer

Breast cancer, as the most common malignant tumor in women, has been studied by researchers from different fields regarding its pathogenesis. The complex interactions between breast cancer cells and immune cells of the adaptive and innate immune systems play a central role in tumor growth, metastasis, and treatment throughout the entire course of the disease [33,138]. In breast cancer progression, serum CHI3L1 levels are closely related to the degree of malignancy of breast cancer, and TAMs also play an important role. The most fatal aspect of breast cancer progression is often the metastasis and invasion of cancer cells in the later stages of the disease. One study showed that, in normal breast tissue, except during the degeneration phase, CHI3L1 is only minimally expressed in mammary ductal epithelial cells, but during lactation, CHI3L1 expression significantly increases, indicating that CHI3L1 mediates the reconstruction and degeneration of mammary epithelial cells [139]. However, overexpression of CHI3L1 in epithelial cells did not cause cell proliferation or carcinogenesis, suggesting that it is not carcinogenic and may participate in the development of tumors through synergistic effects with other carcinogens.

CHI3L1 plays a pathological role in advanced breast cancer rather than in the early stages of the disease, which is consistent with the above research results. Many researchers have found that CHI3L1 protein is specifically secreted by M2 macrophages and induces the expression of MMP-9 in the 76N MECs mammary epithelial cell line by inhibiting E-cadherin, thereby playing a role in the remodeling and degeneration of breast tissue [139]. CAFs are one of the most prominent stromal cells in breast cancer tumors. CAFs promote tumor cell invasion and progression by directly stimulating tumor cells, enhancing angiogenesis, and modifying the extracellular matrix (ECM). In vivo, CAFs promote tumor growth and recruitment by expressing CHI3L1 and activating macrophage MAPK and PI3K signaling pathways to downregulate M1 macrophage-associated factors and promote M2 polarization [33]. This is similar to results obtained by other researchers, specifically results indicating that, in breast cancer progression, CHI3L1 is mainly expressed by M2 macrophages and promotes further metastasis and spread of breast cancer through M2 macrophages.

In addition, compared to other diseases, breast cancer has a higher incidence of metastasis to the lungs in patients with chronic lung disease [140]. The formation and metastasis of tumors are closely related to the generation of neovascularization and the formation of the microenvironment. By studying the role of CHI3L1 in the ‘pre-metastatic’ lungs of breast tumor-bearing mice, it was found that overexpression of CHI3L1 induced the production of vascular growth factors CCL2, CXCL2, and MMP-9 by normal mouse alveolar and interstitial macrophages. In addition to having angiogenic activity, CCL2, CXCL2, and MMP-9 can also act as chemokines to recruit tumor cells, myeloid-derived cells, and macrophages, providing conditions for breast cancer metastasis while promoting further deterioration of the tumor [35,49,130,141].

CHI3L1, as a protein involved in immune suppression in breast cancer, is directly correlated with the downregulation of IFN-γ expression in tumor-bearing mice, suggesting that CHI3L1 may participate in inducing pre-Th2-type responses, thereby reducing the production of Th1-type cytokines [35]. Shibata et al. [142,143] found in a mouse model of allergen-induced chitin treatment that as IFN-γ production increased, the immune response shifted from Th2 to Th1, which is similar to the conclusions drawn by the above researchers and indicates that CHI3L1 primarily affects the progression of breast cancer by influencing macrophage immune mechanisms. A study on shrimp found that the anti-viral miRNA mja-miR-35 in the shrimp body can target the CHI3L1 gene in M2 macrophages in breast cancer mice in a cross-phylum manner, thereby inhibiting breast cancer metastasis and achieving an anti-tumor effect [59]. This interesting study provides a new direction for the treatment of breast cancer.

### 5.2. CHI3L1 and Endometrial Cancer

Endometrial carcinoma (EC) is one of the most common gynecological diseases, with over 417,367 new cases worldwide each year [144]. Chemotherapy and radiotherapy are the most important and common methods in contemporary EC treatment [145]. In the past, CA125 was the only early detection marker accepted clinically for EC, but this marker lacked specificity for EC, as this marker may show levels exceeding the 95th percentile of normal values in a significant proportion of women with benign or malignant diseases [146]. Therefore, finding a more specific marker is critical. CHI3L1 is highly expressed in gynecological tumors such as endometrial cancer [147], cervical cancer [148], and ovarian cancer [149]. A meta-analysis on CHI3L1 included 234 EC patients and 300 normal individuals, showing that 74% of EC patients had elevated CHI3L1 levels, indicating that CHI3L1 is indeed a valuable tumor marker for EC [150]. High CHI3L1 immune reactivity is associated with poor prognosis in endometrial cancer patients, and CHI3L1 also plays a role in angiogenesis and primary and metastatic tumors around tumor blood vessels [151]. As early as 2011, Francescone et al. found that CHI3L1 can promote glioblastoma angiogenesis and radioresistance [152], while Chen et al. found that silencing CHI3L1 can inhibit the VEGF/VEGFR and ERK1/2 signaling pathways, further reducing tumor angiogenesis and inhibiting tumor metastasis [153].

In addition, researchers transfected CHI3L1 siRNA into endometrial cancer cells (HEC-1A), significantly inhibiting CHI3L1 expression and changing HEC-1A migration and invasion potential, suggesting that combining CHI3L1 siRNA with other postoperative chemotherapy may provide more effective treatment for EC patients, effectively preventing postoperative recurrence [154]. Similar results were obtained in an experiment where lentiviral siRNA transfected into HEC-1A and THP-1 cells significantly inhibited the expression of two inflammatory factors, IL-8 and MMP-9, suggesting that macrophages play an important immunomodulatory role in EC progression through CHI3L1 gene silencing and its effects on inflammatory factors.

Inflammation and TAMs play important roles in many cancers, and their relationship with disease progression is close. The relationship between CHI3L1, inflammatory factors, and TAMs may be a potential therapeutic relationship chain, as the close connection between CHI3L1 and TAMs has been found in many other diseases. Similar to breast cancer, CAFs play an important role in regulating the tumor microenvironment in EC, mainly by promoting tumor growth and metastasis through exosomes [155,156]. Researchers have found that the level of NEAT1 released by CAFs in EC is elevated, and NEAT1 promotes tumorigenicity in vivo by regulating the miR-26a/b-5p–STAT3–CHI3L1 axis [157]. Targeting CHI3L1 as a therapeutic target for endometrial cancer is feasible. A common CHI3L1 inhibitor, CHI3L1 neutralizing antibody, has been shown to significantly inhibit angiogenesis and tumor growth [158,159] and promote tumor cell death in various diseases [60]. Based on the important regulatory function of CHI3L1 in EC progression, the screening and use of its related inhibitors are also extremely important. Further research is needed to understand how CHI3L1 participates in the regulation of EC.

### 5.3. CHI3L1 and Colorectal Cancer

In colorectal cancer (CRC), there is a significant expression of CHI3L1 in serum. The overexpression of CHI3L1 alters the tumor microenvironment, thereby enhancing the metastatic potential and sensitivity to cetuximab in CRC [160,161,162]. Evaluation and analysis of various biomarkers’ clinical potential in colorectal cancer patients have revealed that CHI3L1 demonstrates a higher diagnostic value compared to mature miRNA levels in serum (miRNA-576-3p, miRNA-613) and serum NDRG2 levels. In CRC patients, CHI3L1 exhibits an area under the curve (AUC) of 0.97, a specificity of 91.7%, a sensitivity of 96%, and a *p*-value of 0.0001 [163]. Moreover, following liver resection surgery in CRC patients, it has been observed that some patients experience early recurrence. There is a correlation between high levels of CHI3L1 and IL-6 in serum and shorter overall survival. This suggests that monitoring CHI3L1 levels is particularly beneficial for postoperative detection and treatment of CRC patients [164,165].

Early detection of cancer can significantly improve treatment outcomes and patient survival rates, while using CHI3L1 alone as a diagnostic marker for early detection of CRC is not accurate. Therefore, for early diagnosis of colorectal cancer, the combination of CHI3L1 with other markers is usually required to improve accuracy. Based on this, a team has proposed the combined use of stanniocalcin 2 (STC2) and CHI3L1 as a detection method. The combined use of both markers does not seem to alter the predictive performance of STC2 when used alone but provides a new combination for marker utilization [166,167].

CHI3L1 is upregulated in CRC progression and plays a significant role in promoting macrophage infiltration, angiogenesis, and the secretion of IL-8 and MCP-1 [34]. In CRC tumor tissue, CHI3L1 is associated with the expression of MMP-8, IL17A, and PD-L1, thereby influencing the tumor microenvironment [168]. Kawada et al. discovered that CHI3L1 in CRC patients primarily originates from human colorectal cancer cells rather than inflammatory cells. Additionally, CHI3L1 enhances the secretion of IL-8 and MCP-1 in colon cancer cells through signaling mediators such as ERK and JNK [34]. These findings suggest that CHI3L1 not only promotes tumor growth by stimulating cancer cell proliferation but also facilitates it through macrophage recruitment and angiogenesis. Elevated levels of CHI3L1 can increase the sensitivity of cetuximab by downregulating p53 and upregulating EGFR, thereby promoting colorectal cancer cell proliferation. These insights provide therapeutic guidance for the use of cetuximab [34,162,165,169].

Common genetic mutations often increase the risk of colorectal cancer (CRC). The low-frequency coding variant rs3768 in SMAD7 has been found to increase the risk of CRC in the Chinese Han population, while the intronic variant rs4464148 may also affect the prognosis of CRC patients, which indicates the significant importance of the SMAD7 gene [170]. The CHI3L1 gene encodes a protein that regulates the production of TGF-β, and SMAD7, as the most important negative regulator of TGF-β1 activity, is also regulated by it. This genetic explanation sheds light on why CHI3L1 can contribute to the progression of CRC [171].

Unfortunately, despite the early discovery of macrophage infiltration and functional polarization in CRC, the physiological connection between the two remains elusive. Recent studies have focused on the impact of CHI3L1 on the expression of downstream secretory factors in macrophages, but there is no direct evidence demonstrating how they collaboratively promote CRC progression. In an in situ colon cancer model, treatment with CHI3L1 antibodies significantly reduced the growth of primary tumors and suppressed tumor metastasis to the liver [53]. Additionally, the CD31 signal was further reduced in the antibody-treated group, indicating the involvement of CHI3L1 in CD31-mediated disease physiology. These findings provide further hints about the relationship between CHI3L1 and macrophages in CRC. However, the specific mechanisms still require further elucidation by future researchers.

## 6. Conclusions

As a member of the chitinase protein family, CHI3L1 (YKL-40) has been found to be significantly elevated in various diseases, such as liver fibrosis, endometrial cancer, and asthma. In recent years, with further research, similar trends have been found in other diseases, such as esophageal cancer, bladder cancer, allergies, and respiratory diseases. This highlights the importance and necessity of CHI3L1 as a biomarker or therapeutic target.

There are many methods and developments for treating various diseases and cancers including, but not limited to, photodynamic therapy, immunotherapy, radiotherapy, and gene therapy. Immune therapy has many different applications, and immune checkpoint inhibitors play an important role in regulating immune cell expression and modulating immune activation levels. In recent years, research on immune checkpoint blockade (ICB) has made significant progress regarding cancer treatment. ICB uses therapeutic antibodies to disrupt poorly functioning immune checkpoints and activate pre-existing immune responses to achieve therapeutic or interference effects [172]. Antibodies targeting cytotoxic T-lymphocyte antigen 4 (CTLA4) or PD-1–PD-L1 have been approved for use in various cancers. While targeting other potential immune checkpoints, small molecules or antibodies that disrupt negative regulation between tumor cells and T cells or between bone marrow cells and T cells, such as LAG3, TIGIT, TIM3, B7H3, CD39, CD73, and Adenosine A2A receptors, are in clinical trials [172].

Of course, the effect of immune checkpoint blockade is also affected by individual patient factors (age, gender, genetic diversity, etc.), environmental factors, and inherent factors in tumor stroma [173,174,175]. Previous studies have shown that CHI3L1 levels in the bloodstream are significantly elevated in the progression of multiple tumors [22,76,176]. PD-L1, as a ligand–protein pair involved in immune suppression, has been observed to have a close association with CHI3L1 [169]. The expression of CHI3L1 is positively correlated with the expression of immune checkpoints such as CD274 (PD-L1) and HAVCR2 (LAG3) [52]. Knocking down CHI3L1 can regulate cell cycle inhibition, promote cancer cell apoptosis, and enhance the pro-apoptotic effect of anti-PD-L1 antibodies in vitro and in vivo in diffuse large B-cell lymphoma (DLBCL) [177]. This is consistent with another study that demonstrated that blocking CHI3L1 enhances the therapeutic efficacy of anti-PD-L1 treatment [178]. Ma et al. found that PD-L1 was spread in a CHI3L1-dependent manner, meaning that CHI3L1 stimulates CD68 macrophages to produce PD-L1 through its receptor IL-13Rα2. They developed and validated a bispecific antibody—FRG (CHI3L1 antibody) × anti-PD-1, which has significantly better therapeutic effects than the use of single antibodies or the simple addition of two antibodies, with broad application prospects [60]. HA potentially promotes macrophage recruitment and M1 polarization through the IL-2/CHI1L3 and TGF-b/CHI1L3 axes, while also regulating the expression of PD-L1 [44]. Another study has demonstrated that CHI3L1 derived from M2 macrophages induces the expression and secretion of growth differentiation factor 15 (GDF15). This, in turn, coordinates the upregulation of PD-L1 through the activation of PI3K, AKT, and/or ERK pathways [179]. These findings support the close association between macrophage polarization, CHI3L1, and PD-L1, consistent with the relationship between macrophages and CHI3L1 mentioned earlier. However, given the complexity and diversity of diseases, further investigations are needed to elucidate the specific physiological mechanisms underlying the interactions between macrophages, CHI3L1, and PD-L1 in different diseases.

Various data indicate that CHI3L1 is a key regulatory factor in numerous diseases. It participates in macrophage infiltration, M1/2 differentiation, Th1/2 activation, and the release of proinflammatory and anti-inflammatory factors through classical signaling pathways such as JNK, ERK, MAPK, PI3K, STA3/6, and others. It also participates in influencing disease progression, tumor development and metastasis, and vascular genesis (shown in Figure 4). This article does not summarize all diseases that CHI3L1 participates in but lists in-depth studies on several diseases, including atherosclerosis, breast cancer, endometrial cancer, colorectal cancer, Alzheimer’s disease, and liver disease. Throughout the review process, it was found that there is a close correlation between CHI3L1 and macrophages.

Most importantly, the use of CHI3L1 antibodies as a treatment direction seems feasible and effective based on laboratory results. However, since most of the research is concentrated on mouse models, the actual use of antibodies still requires a long period of clinical research evaluation.

## Figures and Tables

**Figure 1 ijms-24-16149-f001:**
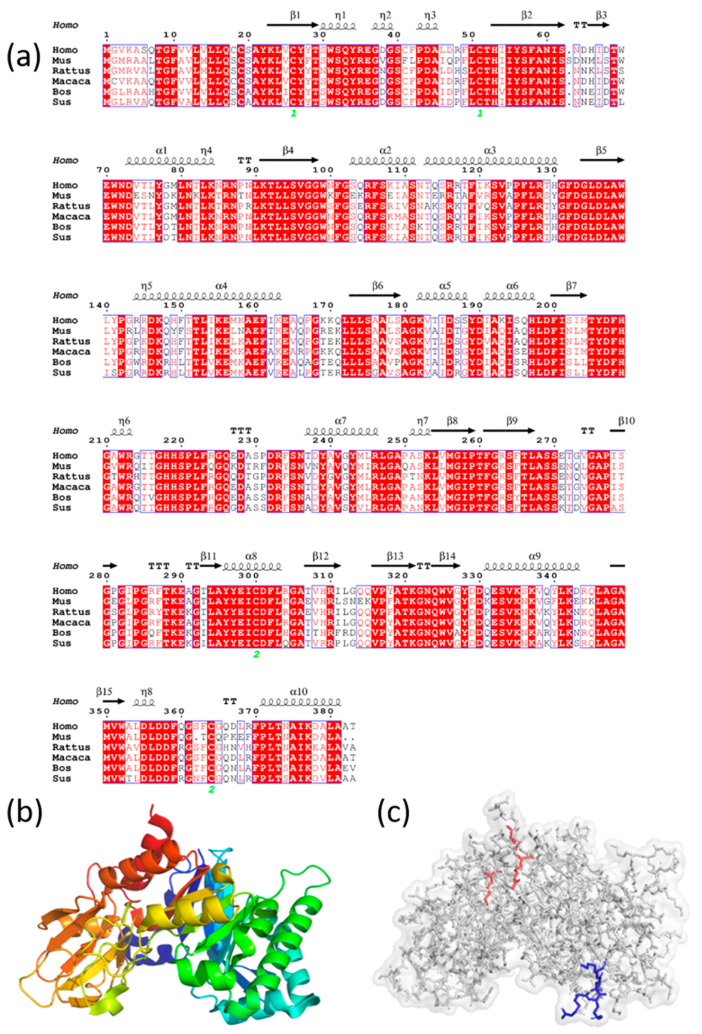
Homologous sequence alignment, crystal structure, and heparin-binding site of CHI3L1. (**a**) Homologous sequence alignment of CHI3L1 from six different species, including Homo sapiens, Mus musculus, Rattus norvegicus, Bos Taurus, Sus scrofa, and Macaca mulatta; (**b**) Crystal structure of CHI3L1 (PDB code: 1NWR); (**c**) Proposed amino acids (R143, R144, D145, K146) in heparin-binding motif shown in blue and amino acids (K337, K342, R344) in actual binding sites shown in red.

**Figure 2 ijms-24-16149-f002:**
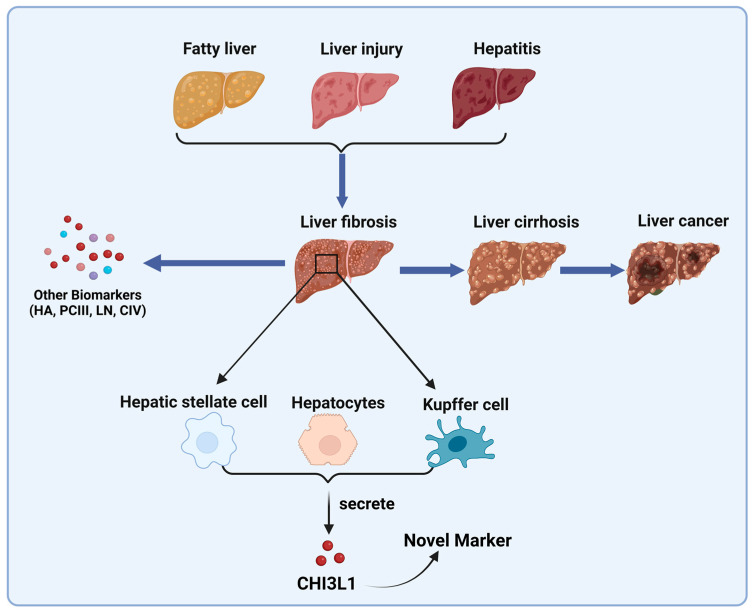
CHI3L1 as a diagnostic and staging biomarker in liver fibrosis progression. In the progression of liver fibrosis, CHI3L1 is secreted and expressed by hepatocytes, hepatic stellate cells, and Kupffer cells in the liver, thus making it a promising diagnostic and staging biomarker for liver fibrosis.

**Figure 3 ijms-24-16149-f003:**
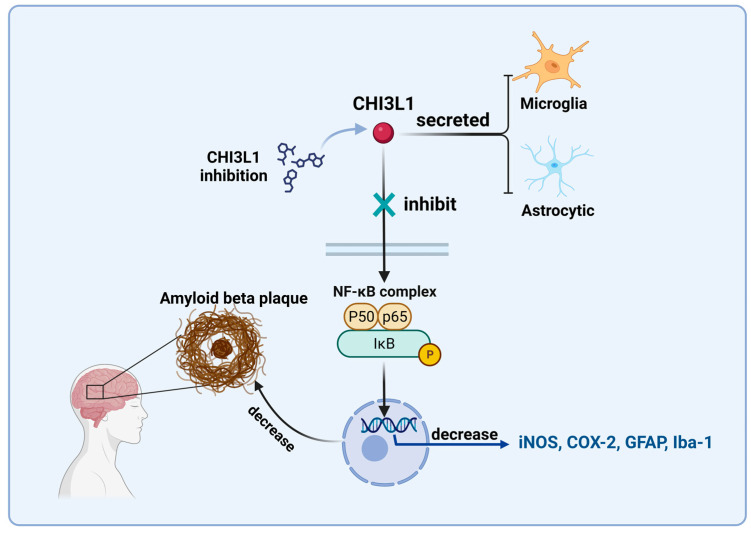
Inhibition of the NF-κB signaling pathway by small-molecule inhibitors: In AD progression, CHI3L1 is primarily secreted by microglia and astrocytes. Small molecule inhibitors reduce the expression of genes such as iNOS, and COX-2 by inhibiting the activation of the NF-κB signaling pathway, slowing down the deposition of amyloid plaques, and improving disease progression.

**Figure 4 ijms-24-16149-f004:**
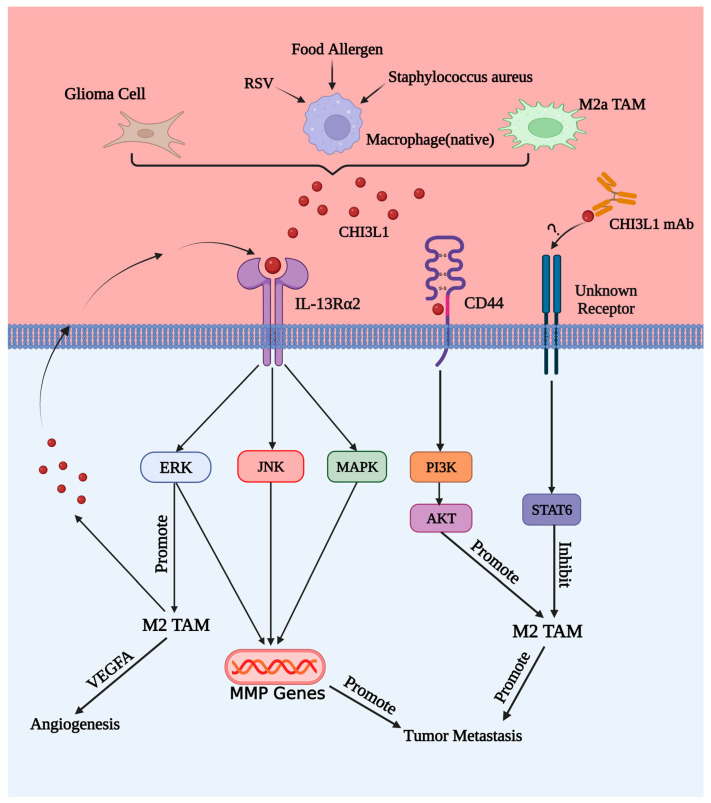
Activation of signaling pathways and biological effects induced by CHI3L1 interaction with different receptors: The interaction of CHI3L1 from different sources with various receptor sets activates signaling pathways such as JNK, ERK, and PI3K, promoting the polarization and infiltration of M2 macrophages. This, in turn, leads to angiogenesis and tumor metastasis.

**Table 1 ijms-24-16149-t001:** The relationship between CHI3L1 and macrophage polarization in different diseases.

Disease Type	CHI3L1 Change	Research Objects	Macrophage Polarization	Signaling Pathway	CHI3L1 Receptor	CHI3L1 Function in Disease	Year and Country	Reference
Glioblastoma	Increased	Cell, Mice	M2 infiltration	CHI3L1/Gal3-PI3K/AKT/mTOR	Gal3	Promotes macrophage-mediated immune suppression	2021, USA	[48]
Breast cancer, gastric cancer	Increased	Cell, Mice	M2 infiltration	ERK, JNK, MAPK	IL-13Rα2	Promotes cancer metastasis	2017, China	[49]
IDD	Increased	Cells, Human Tissues	M2a infiltration	ERK, JNK	IL-13Rα2	Promotes ECM degradation	2021, China	[50]
ESCA	Increased	Cells, Mice, Human tissues	M2-like polarization	TGF-β	unknown	Stimulates M2 gene expression	2021, China	[51]
Staphylococcus aureus infection	Increased	Cell	M1-like polarization	ERK, JNK	unknown	Macrophage polarization	2017, China	[47]
Glioma	Increased	Cell, Mice, Human serum and samples	M2-like polarization	NF-κB	ACTN4, NFKB1	Promotes tumor cell proliferation and survival	2022, China	[52]
TME	Increased	Cell, Mice, Human serum and samples	M2-like polarization	AKT, β-catenin, NF-κB	Unknown	Promotes tumor growth and metastasis	2022, China	[53]
Colorectal Cancer	Increased	Cell, Mice, Human serum and samples	Macrophage infiltration	ERK, JNK	Unknown	Promotes the chemotaxis of macrophages and angiogenesis	2012, Japan	[45]
nAMD	Increased	Cell, Mice, Human serum	M2-like polarization	ERK	IL13-Ra2	Angiogenesis	2019, China	[42]
Food Allergy	Increased	Cell, Mice, Human Serum	M2-like polarization	ERK, AKT	IL13-Ra2	Th2-associated inflammation and M2 macrophage polarization	2020, Korea	[43]
AD	Increased	Cell, Mice, Human Tissues	M2 macrophage activation	Unknown	Unknown	Affects the development of AD	2019, Korea	[45]
Breast Tumor	Increased	Cell, Mice	Reprogramming to an M2-like phenotype	MAPK, PI3K	Unknown	Promotes tumor progression	2017, Israel	[33]

Abbreviations: IDD, Intervertebral Disc degeneration; ESCA, esophageal cancer; TME, Immunosuppressive Tumor Microenvironment; nAMD, Neovascular Age-related Macular Degeneration; AD, Atopic Dermatitis.

**Table 2 ijms-24-16149-t002:** Overview of CHI3L1 inhibitors.

Type	Compound	Related Diseases	Mechanism	Binding Area	Impact on CHI3L1	Key Protein/Cell	Related Pathway	Results	Year and Country	Reference
Small molecular	K284-6111	Lung cancer	Prevents the binding of CHI3L1 to receptor IL-13Rα2	chitin-binding domain (CBD)	Inhibition	IL-13Rα2	JNK, AKT, AP-1	Prevents lung cancer cell metastasis and growth.	2022, Korea	[54]
	K284-6111	Alzheimer’s disease	Suppresses p50 and p65 translocation into the nucleus and phosphorylation of IκB in vivo and in vitro	chitin-binding domain (CBD)	Inhibition	NF-κB	NF-κB, IκB	Anti-amyloidogenic and anti-neuroinflammatory effects with improving neuronal survival and memory deficiency	2018, Korea	[55]
	G721-0282	Osteosarcoma (OS)	Induces the inactivation of mitogen-activated protein kinases (MAPKs) with a decrease in the phosphorylation of Src and STAT3 in OS cells	unknown	Inhibition	STAT3	STAT3	Inhibits the proliferation of OS cells and induces apoptosis	2020, Korea	[56]
	G721-0282	Chronic unpredictable mild stress (CUMS)	Regulates IGFBP3-mediated neuroinflammation via inhibition of CHI3L1	unknown	Inhibition	IGFBP3	NF-κB	Lower CUMS-induced anxiety-like behaviors	2022, Korea	[57]
Natural molecular	Ebractenoid F	Lung cancer	significantly decreases phosphorylated AKT expression	unknown	Inhibition	AKT	AKT	Inhibits lung cancer cell growth	2022, Korea	[58]
MicroRNAs (miRNA)	mja-miR-35	Brest cancer	silences CHI3L1 gene expression in a cross-phylum manner	unknown	Inhibition	unknown	unknown	Inhibits breast cancer metastasis	2018, China	[59]
	miR-24	Staphylococcus aureus	significantly inhibits the M1 phenotype; increases M2 phenotype	unknown	Inhibition	Macrophage	MAPK	Downregulates the expression level of CHI3L1 and regulates the MAPK pathway in S. aureus- induced macrophages	2017, China	[47]
Bispecific antibody	FRGxPD-1	Melanoma	simultaneously induces CD8+ CTL differentiation and tumor cytotoxicity	unknown	Inhibition	CD8+ cytotoxic T cells	Wnt/β-catenin	Generates an additive antitumor response	2020, USA	[60]
Antibody	polyclonal CHI3L1-neutralizing antibodies (nCHI3L1 Abs)	Lung, pancreas, and colon tumor models	enhances CD8+ T-cell cytotoxicity and attenuates Treg and M2-type macrophage polarization to induce anti-tumor properties in vitro	unknown	Inhibition	Rab37, M2	AKT, NF-κB, β-catenin	Reduces tumor growth/metastases and elicits an immunostimulatory TME	2022, China	[53]

## Data Availability

Data sharing not applicable.

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
