# Peer review of "Potential Roles and Future Perspectives of Chitinase 3-like 1 in Macrophage Polarization and the Development of Diseases"

_ijms, 2023, doi:10.3390/ijms242216149_

Round 1

Reviewer 1 Report

Comments and Suggestions for Authors

In the present manuscript, Zhao et al. comprehensively summarize the role of CHI3L1 in the development of various diseases and the therapeutic targeting of CHI3L1. The manuscript is well-organized, covering a vast area, and full of important information for the readers in the field. I have a bunch of minor concerns which should be addressed before the final acceptance of the manuscript. The comments are shown below:

1. Abstract, line 8: remove diseases after cancer.

2. Introduction, line 3: remove 's' from the 'fields'.

3. Use abbreviation CLPs instead of chitinase-like proteins.

4. Introduction, line 6: put are after CLPs.

5. Page 2, Introduction, first paragraph last line: put 'of CHI3L1' after recognition.

6. Page 2, Introduction, second paragraph, line 3: Fig. 2. a). should be Fig. 1. a).

7. Page 2, Introduction, third paragraph, line 6: put reference after SPP.

8. Page 2, Introduction, third paragraph, line 7: put 'moreover' instead of 'furthermore'.

9. Page 2, Introduction, third paragraph, line 8: put full name for SC2.

10. Page 2, Introduction, third paragraph, last 2 lines: put full form of VOC.

11. Page 2, Introduction, last paragraph, line 5: put 'the' instead of 'this' after 'based on this'.

12. Page 3, first line: write 'activities' instead of 'capabilities'.

13. Page 3, first paragraph, line 7: put reference after the statement '.... tissue homeostasis'.

14. Page 3, second paragraph, line 7: should be 'can be transformed'.

15. Page 3, last paragraph, line 4: it should be 'understanding'.

16. Page 4, second paragraph, line 3: 'obtained' should be 'observed'.

17. Page 4, fourth paragraph, line 11: should be 'extent is advantageous'. Page

18. Page 4, fourth paragraph, line 12: omit 'us' after 'bring'.

19. Put abbreviations used in Table 1.

20. Page 12, first paragraph: put reference for the first sentence.

21. Page 12, last paragraph, last line: remove 'most' before 'highly'.

22. Page 13, first paragraph, line 8: it should be 'compared to'.

23. Section 4.2 heading: 'liver-related diseases' should be 'liver diseases'.

24. Section 4.2, second paragraph, line 5: 'compared with' should be 'compared to'.

25. Reference 100 sentence is too long to follow, shorten it.

26. Section 4.2, last paragraph, line 11: remove 'research' before 'results'.

27. Page 14, first paragraph, line 9: put 'macrophages' instead of symbol for phage.

28. Page 15: use the abbreviation 'CSF' for 'cerebrospinal fluid'.

29. Page 16, third paragraph, line 1: is it 'glia' or 'glial cells'?

30. Page 32, first paragraph, line 5: Add the reference 'PMID: 37579880' along with the existing reference [142].

31. Page 33, second paragraph, line 5: it should be 'similar results were obtained'.         

Comments on the Quality of English Language

A few grammatical errors remain, I suggested some.

Author Response

Please see the attachment, thanks.

Reviewer 2 Report

Comments and Suggestions for Authors

The manuscript named “Potential roles and future perspectives of Chitinase 3-like 1 in macrophage polarization and the development of diseases” by Zhao et al. is dedicated to the important chitinase-like protein that has been demonstrated may cleave chitin to prevent invasion by chitin-carrying pathogens.  Chitin is an important structural polysaccharide, that supports and organizes extracellular matrices in a variety of taxonomic groups including bacteria, fungi, protists, and animals, thus the detection and destruction of chitin is important for immunity in case of invasion. The levels of Chitinase 3-like 1 (CHI3L1) protein are significantly elevated in inflammatory diseases, but also in cancer, and degenerative diseases, and are closely associated with patient survival and poor prognosis.

The review is well written in good English and meticulously reviews Chitinase 3-like 1 protein function and the effects of its activity or inhibition on many health complications it is involved.  The manuscript does not summarize all diseases that CHI3L1 participates in but lists in-depth studies on several diseases, including atherosclerosis, breast cancer, endometrial cancer, colorectal cancer, ‘Alzheimer’s disease, and liver disease. Throughout the review process, it was found that there is a close correlation between CHI3L1 and macrophages.

This reviewer liked the author's work, as I said, it is meticulous and clear. The only remark I want to make is that the authors are probably afraid to make more generalizations. Reading their work, it is possible to suggest that Chitinase 3-like 1 is necessary for the detection of fungal infections. For example, it is known that invasive fungal infections are very common and have become a leading cause of morbidity and mortality in cancer patients (Angarone M. Fungal infections in cancer patients. Cancer Treat Res. 2014;161:129-55. doi: 10.1007/978-3-319-04220-6_4). Also, fungal macromolecules can be detected in the brain of Alzheimer's disease patients,  representing a risk factor or may contribute to the etiological cause of Alzheimer's disease ( Alonso R, Pisa D, Marina AI, Morato E, Rábano A, Carrasco L. Fungal infection in patients with Alzheimer's disease. J Alzheimers Dis. 2014;41(1):301-11. doi: 10.3233/JAD-132681). One may suggest that CHI3L1 elevated in these health complications may represent the response to fungal infection as well, augmenting a vicious cycle of inflammation leading to beta-amyloid production and accumulation in affected tissues (Inyushin M, Zayas-Santiago A, Rojas L, Kucheryavykh L. On the Role of Platelet-Generated Amyloid Beta Peptides in Certain Amyloidosis Health Complications. Front Immunol. 2020 Oct 2;11:571083. doi: 10.3389/fimmu.2020.571083).

It would be interesting if authors would add a short argument on this point in their work. After these small changes, I suggest the manuscript can be accepted.

Author Response

Please see the attachment, thanks.
